# Glycosylation and Its Role in Immune Checkpoint Proteins: From Molecular Mechanisms to Clinical Implications

**DOI:** 10.3390/biomedicines12071446

**Published:** 2024-06-28

**Authors:** Jingyi Liu, Ximo Xu, Hao Zhong, Mengqin Yu, Naijipu Abuduaini, Sen Zhang, Xiao Yang, Bo Feng

**Affiliations:** Department of General Surgery, Ruijin Hospital, Shanghai Jiao Tong University School of Medicine, Shanghai 200000, China; liujingyi0907@163.com (J.L.); xxm970603@163.com (X.X.); wmuzh2@163.com (H.Z.); yumengqin@sjtu.edu.cn (M.Y.); najip199471@sjtu.edu.cn (N.A.); zhangsen6886@163.com (S.Z.); yxrjmis@alumini.sjtu.edu.cn (X.Y.)

**Keywords:** immunotherapy, immune checkpoint, glycosylation, sialylation, PD-L1, siglecs

## Abstract

Immune checkpoint proteins have become recent research hotspots for their vital role in maintaining peripheral immune tolerance and suppressing immune response function in a wide range of tumors. Therefore, investigating the immunomodulatory functions of immune checkpoints and their therapeutic potential for clinical use is of paramount importance. The immune checkpoint blockade (ICB) is an important component of cancer immunotherapy, as it targets inhibitory immune signaling transduction with antagonistic antibodies to restore the host immune response. Anti-programmed cell death-1 (PD-1) and anti-cytotoxic T lymphocyte-associated antigen-4 (CTLA-4) monoclonal antibodies are two main types of widely used ICBs that drastically improve the survival and prognosis of many patients with cancer. Nevertheless, the response rate of most cancer types remains relatively low due to the drug resistance of ICBs, which calls for an in-depth exploration to improve their efficacy. Accumulating evidence suggests that immune checkpoint proteins are glycosylated in forms of N-glycosylation, core fucosylation, or sialylation, which affect multiple biological functions of proteins such as protein biosynthesis, stability, and interaction. In this review, we give a brief introduction to several immune checkpoints and summarize primary molecular mechanisms that modulate protein stability and immunosuppressive function. In addition, newly developed methods targeting glycosylation on immune checkpoints for detection used to stratify patients, as well as small-molecule agents disrupting receptor–ligand interactions to circumvent drug resistance of traditional ICBs, in order to increase the clinical efficacy of immunotherapy strategies of patients with cancer, are also included to provide new insights into scientific research and clinical treatments.

## 1. Introduction

Under normal physiological conditions, immune checkpoints maintain peripheral tolerance by transducing inhibitory signals to suppress T-cell activation, proliferation, and even cytokine production. However, some tumor cells can express immune checkpoint proteins on their cell surface to suppress an antitumor immune response, and thus mediate tumor escape [1]. Therefore, the Food and Drug Administration (FDA) has approved several immune checkpoint blockades (ICBs) to fight against various cancer types. Even though the application of ICBs radically changed the immunotherapy landscape, only a tiny percentage of patients with tumors achieved a satisfying response [2].

Glycosylation is one form of post-translational modification (PTM). Glycan moieties are covalently bonded to proteins or lipids as glycoconjugates by an enzymatic site-specific mechanism. N-linked and O-linked glycosylation are two main types of glycosylation, depending on the linkages they form with different amino acid residues on the polypeptide backbone (Figure 1A). The asparagine (Asn) side chain acceptor, located within the NXT motif (-Asn-X-Ser/Thr-), gets an oligosaccharide-based glycan catalyzed by the oligosaccharyltransferase (OST) complex in the endoplasmic reticulum (ER), where N-linked glycosylation is initiating. Then, the glycoprotein moves to the Golgi apparatus for additional glycosidase- and glycosyltransferase-mediated modifications [3] (Figure 1A). O-linked glycosylation mainly happens in the Golgi apparatus where N-acetylgalactosamine (GalNAc) is attached to serine (Ser) or threonine (Thr) on the polypeptide backbone via O-linkage (Figure 1A). Glycosylation serves a crucial role in regulating many processes in physiological and pathological ways, including protein biosynthesis, stability, and their interactions with other molecules, and this is why glycosylation interferes with various critical cancer cell processes and promotes tumor progression. 

Many immune checkpoints are known to be heavily glycosylated. Since glycosylation helps maintain protein stability and mediate protein–protein interactions, the study of glycosylation regulation of immune checkpoints may help enhance the detection rate for better patient stratification and prediction of therapeutic efficacy [4]. In addition, the further development of glycosylation-specific antibodies can help to promote the response rate of immunotherapy and hopefully, improve the survival of patients with tumors. 

In this review, we briefly introduce several immune checkpoints and summarize primary molecular mechanisms of glycosylation that modulate protein stability and immunosuppressive function. In addition, newly developed methods of detection, as well as small-molecule agents, are also included to provide new insights into scientific research and clinical treatments.

## 2. The Glycosylation of PD-L1

Programmed cell death-ligand 1 (PD-L1) blocks T-cell activation and cytokine production once it binds to programmed cell death-1 (PD-1) (Figure 1B). After eliminating the N-glycan structure by PNGase F, the N-glycosidase, PD-L1’s molecular weight is reduced to 33 kDa from ~45 kDa, suggesting a high level of N-glycosylation in PD-L1. PD-L1 glycosylation occurred on its extracellular domain (ECD) and the four sites for N-glycosylation are N35, N192, N200, and N219 [5] (Figure 1B). PD-L1 glycosylation is critical for its biological functions, especially maintaining its protein stability and mediating interactions with PD-1. 

### 2.1. Glycosylation Regulates PD-L1 Protein Stability

**β-catenin/STT3:** N-glycosyltransferase STT3 is a catalytic subunit of the OST complex, and it catalyzes the transfer of a 14-sugar core glycan from dolichol to the asparagines of substrates, hence starting N-glycosylation [6]. STT3 has two isoforms, STT3A and STT3B, which act on polypeptides successively to increase N-glycosylation efficiency [7]. Hsu et al. discovered that knocking down both STT3 isoforms suppresses epithelial–mesenchymal transition (EMT)-mediated PD-L1 induction in protein levels, reducing its molecular weight to ~33-kDa, which is close to that of unglycosylated PD-L1, driving the conclusion that STT3 induces PD-L1 through protein glycosylation and stability. Then, the authors tried to explore how EMT induces STT3 isoforms. It turned out that β-catenin, an EMT transcription factor, activates the promoters of STT3 isoforms by binding to the transcription factor TCF7L2, therefore inducing STT3 expression and forming the EMT/β-catenin/STT3/PD-L1 signaling axis (Figure 2) [8]. Additionally, it was shown that transforming growth factor β (TGF-β) can upregulate STT3A expression through c-Jun, which improves the glycosylation and stability of PD-L1 [9]. Therefore, agents targeting the β-catenin/STT3/PD-L1 axis and their upstream signals might achieve improved treatment efficacy against PD-L1 glycosylation, especially targeting STT3 directly. 

**Sigma1/FKBP51s**: Sigma1 and the splicing isoform of the FK506-binding protein-51 (FKBP51s) are both chaperones that assist PD-L1 glycosylation [10,11]. Sigma1 primarily associates with glycosylated-PD-L1, and if treated with IPAG (1-(4-Iodophenyl)-3-(2-adamantyl) guanidine), a pharmacological inhibition of Sigma1, PD-L1 expression will be decreased through autophagic degradation and therefore promote T-cell activation [11] (Figure 2). FKBP51s acts as a foldase and a catalyst of PD-L1 in the ER, where it forms a complex with PD-L1. FKBP51s not only catalyzes protein folding but also maintains the stability of glycosylated-PD-L1 [10]. D’Arrigo et al. revealed that by using the chemical agent SAFit, which can specifically block FKBP51s’ catalytic activity [12], significant reductions were observed in PD-L1 expression and PD-L1-induced cell death in peripheral blood mononuclear cells (PBMCs) cocultured with glioma cells [10] (Figure 2). 

**B4GALT1**: Within the β4-galactosyltransferase (B4GALT) family, B4GALT1 is one of the seven members whose duty is to transfer UDP-galactose residues to terminal N-acetylglucosamine residues during glycosylation [13]. In triple-negative breast cancer (TNBC), Zhang et al. identified that B4GALT1 bound by RBMS1 at 3′-UTR stabilizes its mRNA and promotes the glycosylation of PD-L1 and it protects PD-L1 from being ubiquitinated and degraded by proteasome [14] (Figure 2). Similarly, in lung adenocarcinoma, Cui et al. discovered that B4GATL1 promotes tumor immune escape by mediating PD-L1 N-glycosylation to increase its stability at the post-transcriptional level [15]. 

### 2.2. The Glycosylation of PD-L1 Promotes Its Immunosuppressive Function

To ascertain whether the immunosuppressive function is modulated by the glycosylation of PD-L1, Li et al. assessed T-cell response and examined the tumorigenesis of tumor cells. It turned out that in BALB/c animals, non-glycosylated PD-L1 (ngPD-L1) cells grew noticeably more slowly, were more susceptible to activated T-cell-mediated death, and produced increased IL-2 secretion compared to glycosylated PD-L1 (gPD-L1). This conclusion was further supported by the discovery that gPD-L1-induced tumors had fewer activated cytotoxic T-cells in their TILs than ngPD-L1 tumors [16]. Overall, this is likely due to glycosylation’s dual effects of stabilizing PD-L1 and facilitating the interaction between PD-1 and PD-L1, which inhibits the immune functions of T-cells in the tumor microenvironment. For example, Liu et al. found that in B-cell non-Hodgkin’s lymphoma (NHL), glycosyltransferase 1 domain-containing 1 (GLT1D1) is essential for the transfer of N-linked glycans to PD-L1 (Figure 2), which promotes tumor growth by facilitating tumor immunosuppressive function and leads to a poor prognosis [17]. Conversely, GLT1D1’s downregulation may result in a reduction in glycosylated PD-L1, which would reactivate cytotoxic T-cells against lymphoma cells [17]. 

### 2.3. Relationships between PD-L1 Glycosylation and Other Post-Translational Modifications

Apart from glycosylation, other PTMs of PD-L1 cooperate to control biofunctions and PD-L1 degradation. Among them, glycosylation is tightly intertwined with phosphorylation, ubiquitination, and following degradation. 

Glycogen synthase kinase 3β (GSK3β) is a Ser/Thr protein kinase that mediates phosphorylation to facilitate ubiquitin E3 ligase recognition [18]. Non-glycosylated PD-L1 was phosphorylated by GSK3β at the T180 and S184 sites of its ECD [5]. The glycosylation of certain residues can counteract the interaction between PD-L1 and GSK3β, thus suppressing GSK3β-β-TRCP-mediated PD-L1 polyubiquitination and 26S proteasome degradation, promoting PD-L1 stability [5] (Figure 2). Additionally, epidermal growth factor (EGF) is known to inhibit GSK3β activity, and Li et al. examined that PD-L1 expression at the post-translational level can be dramatically induced by EGF in combination with several EGFR ligands [5] (Figure 2). According to these findings, Li et al. further explored that gefitinib, an EGFR inhibitor, can inhibit PD-L1 binding to other T-cell receptors in addition to lowering PD-L1 expression and can reduce EGFR-overexpressing tumor cell survivals, which makes it an excellent choice for combination therapy with anti-PD-1 antibodies [5]. 

The direct phosphorylation of PD-L1 at S195 by metformin-activated AMP-activated protein kinase (AMPK) modifies the glycan structure of PD-L1, inducing aberrant glycosylation with mannose-rich structures responsible for excessive ER mannose trimming, and subsequently leads to ER accumulation and ER-associated protein degradation (ERAD) [19] (Figure 2). Moreover, to preserve protein stability in the ER, non-glycosylated PD-L1 is phosphorylated by the IL-6/JAK1 pathway at the Y112 residue and increases PD-L1’s interaction with STT3A [20] (Figure 2). Since increased levels of IL-6 in the tumor site or plasma are associated with a worse prognosis for patients with advanced hepatocellular carcinoma (HCC), inhibiting the IL-6 pathway may enhance current HCC therapeutic approaches [20] (Figure 2).

### 2.4. Glycosylation in Clinical Diagnosis and PD-L1 Detection

The PD-L1 blockade drastically altered clinical cancer treatment and greatly boosted patients with cancer’s survival [21]. In addition to PD-L1-positive patients, some PD-L1-negative patients also respond favorably to immunotherapy, suggesting that the pathological evaluation of PD-L1 level is not an accurate predictor of the effectiveness of anti-PD-L1 therapy [22,23,24]. Therefore, to reduce the disparities between the patient’s response and the PD-L1 level, a more precise biomarker is needed. Glycosylation has the potential to be a novel biomarker for PD-L1 detection and patient stratification [25].

Lee et al. postulated that glycan moieties on PD-L1 form a steric hindrance for antibody recognition and binding, thus leading to inaccurate immunohistochemical (IHC) readouts. Therefore, they conducted a retrospective study using PNGase F to attain the deglycosylation of cancer cells. It turns out that removing glycan moieties largely improves PD-L1 detection sensitivity by anti-PD-L1 antibodies. The number of patients who had positive PD-L1 IHC staining rose significantly after deglycosylation [4]. Therefore, sample deglycosylation enhanced the evaluation of PD-L1 expression and the capacity to forecast the clinical result of anti-PD-L1 therapy [4]. For example, a sizeable population of patients with non-small cell lung cancer (NSCLC) who ought to have qualified for ICB treatment were excluded only because of false-negative IHC readouts. After deglycosylation, patients exhibited a profound increase in PD-L1 tumor proportion score (TPS), making them suitable for immune checkpoint treatment [4].

Morales-Betanzos et al. came to the same conclusion through a comparison between mass spectrometry (MS) and IHC. They employed MS to analyze PD-L1 glycosylation in 22 specimens of patients with melanoma treated with ICBs. They not only confirmed PD-L1glycosylation, but also identified that consistent N-glycosylation at N192 has high mannose structures. Therefore, the abundance of the glycoforms can be estimated and normalized to the quantified amount of PD-L1 as a parameter to compare the fractional N192 glycosylation of PD-L1 in the samples. However, some samples exhibited a high level of PD-L1 abundance and N192 glycosylation through MS, but a low level of PD-L1 was detected by IHC. This discordance implied that PD-L1 glycosylation might interfere with the recognition of IHC [26].

In conclusion, by conducting deglycosylation treatment before IHC staining to remove steric hindrance, PD-L1 becomes a more precise biomarker for patient stratification and clinical response prediction for anti-PD-L1 therapy.

### 2.5. Agents Targeting Glycosylated PD-L1

Based on the diverse impacts of glycosylation on the stabilization of PD-L1 discussed above, much emphasis should be put on the exploration of specific small-molecule agents targeting glycosylated PD-L1, so that the efficacy of immunotherapy can be largely improved. Pu et al. put forth the “non-glycosylated PD-L1” concept for tumor immunotherapy by blocking related enzymes and specific molecules or applying small molecule PD-L1 inhibitors [25].

Li et al. selected STM004 and STM108 as two glycosylation-specific PD-L1 antibodies. STM108 recognized the N192 and N200 glycosylation sites, whereas STM004 recognized the N35 glycosylation site (Figure 2). Both of them exhibited potent effects on the reduction in tumor size and elevation of cytotoxic T-cell activity. Furthermore, STM108 can specifically recognize and bind to a poly-N-acetyllactosamine (polyLacNAc) moiety synthesized by β-1,3-N-acetylglucosaminyltransferase-3 (B3GNT3) [15]. Although PD-L1 is broadly expressed in normal and tumor cells [27], PD-L1 in tumor cells has upregulation in expression and site-specific aberrant glycan modifications that distinguish tumor cells from normal ones [28], and this decides its malignancy selectivity, making it an excellent choice for drug conjugation. The STM108 antibody is a perfect fit for ADC because it causes PD-L1 internalization to lysosomes and subsequent degradation [15]. Therefore, an STM108 antibody conjugated with the effective anti-mitotic medication monomethyl auristatin E (MMAE) called STM108-ADC was produced, which markedly induced tumor regression [29]. STM108-ADC proved to be extremely effective and quite safe, indicating a potentially risk-free clinical use [15].

Activating FcγR is the mechanism by which antibodies with Fc-mediated effector function trigger antibody-dependent cellular cytotoxicity (ADCC) against TILs that express PD-L1 [30]. The Fc region plays an essential role in the effectiveness of anti-PD-L1 antibodies by mediating greater antitumor activity through isotypes that have a higher affinity for activating FcγRIIIa in humans and FcγRIV in mice, due to species–specific differences [31]. Since human IgG antibodies normally include two conserved N-linked oligosaccharides [32], the increasing affinity for FcγRIIIa is usually achieved by removing only the core fucose from the N-glycans [33]. Therefore, Goletz et al. produced two anti-PD-L1 antibody variants with varying degrees of core fucosylation, and αPDL1_GE_, which had minimized core fucosylation, showed enhanced binding affinity to FcγRIIIa [34]. Additionally, αPDL1_GE_ generated increased ADCC against PD-L1^+^ cancer cells without undesired ADCC against B cells and monocytes. It also dramatically boosted CD8^+^T-cell activation, which translated into stronger cytotoxicity against cancer cells [34]. Taken together, glyco-engineered anti-PD-L1 antibodies can hopefully improve the therapeutic benefits of anti-PD-L1 immunotherapy.

2-deoxyglucose (2-DG) can act as a glucose analog to change the glycosylation composition of PD-L1 (Figure 2). As a result, 2-DG can not only reduce cell surface PD-L1 expression by downregulating the glycosylation of PD-L1, but can also deglycosylate the PARPi-generated PD-L1 protein in TNBC to induce the cancer cells more susceptible to T-cell-mediated death. This study offers a strong scientific basis for additional investigation into the use of 2-DG and PARPi together in the treatment of TNBC [35]. As discussed above, EGF signaling stabilizes the PD-L1 protein by increasing glycosylation; therefore, 2-DG and gefitinib combination treatment can result in improved antitumor immunity. Given that 2-DG/gefitinib-induced PD-L1 deglycosylation might not reduce the therapeutic benefit of 4-1BB agonist antibodies, Kim et al. clarified that combined 2-DG/gefitinib and an agonist agent of 4-1BB, which belongs to the TNF receptor superfamily, can jointly boost the antitumor immune response in TNBC mouse models [36]. Taken together, 2-DG can be an ideal candidate for combination therapy with several agents.

Resveratrol (RSV) can directly inhibit PD-L1 glycosylation catalytic enzymes, which regulate the N-linked glycan trimming of PD-L1. This helps the ER retain an aberrantly glycosylated, mannose-rich version of PD-L1 [37] (Figure 2). First shown by Verdura et al., RSV inhibits cancer cells’ ability to evade immunosuppression by directly rupturing N-linked glycosylation and encouraging PD-L1 dimerization. This interferes with PD-L1’s proper localization to the cell membrane, eliminates the interaction with PD-1, and ultimately makes tumor cells more vulnerable to T-cell killing [37]. Hopefully, RSV might shed light on fresh strategies to manipulate immunomodulatory molecules by targeting immune checkpoints to restore T-cell function.

## 3. The Glycosylation of PD-1

PD-1 is located on the membrane of activated T-cells to hinder CD28/T-cell receptor (TCR) signaling by engaging with PD-L1, which leads to T-cell exhaustion and immunosuppression [27]. PD-1 is also heavily N-glycosylated in eukaryotic cells. According to Tan et al., the IgV domain of the ECD of PD-1 interacts with the IgV domain of its ligands, PD-L1 or PD-L2, and has four possible N-glycosylation sites (N49, N58, N74, and N116) [38,39]. The glycosylation of PD-1 not only mediates immunosuppression via PD-1/PD-L1 interaction, but also affects the activity of PD-1-specific monoclonal antibodies (mAbs) [40,41].

### 3.1. Glycosylation Regulates PD-1 Protein Expression and Stability

One distinctive type of N-linked glycosylation is fucosylation, and it is catalyzed by a range of specific fucosyltransferases (FUTs) [42]. Fucosylation alters the glycosylation of signaling receptors, which is crucial for tumor growth, invasiveness, and resistance to chemotherapy [43].

Attaching an α-1,6 fucose to the innermost N-acetylglucosamine (GlcNAc) residues in N-glycans by FUT8 is known as core fucosylation. Okada et al. identified that all four N-glycosylation sites of PD-1 are highly core fucosylated. Among these, the N49 and N74 residues are required for the proper expression and function of PD-1. The interface between PD-1 and PD-L1 is located close to the N74 residue [44]. Core fucosylation and its catalytic enzyme FUT8 have been demonstrated to be associated with the regulation of PD-1 stability and expression in multiple ways [45]. Okada et al. found that FUT8 expression levels are positively linked with PD-1 by controlling its cell surface expression and functional location. Additionally, FUT8 and subsequent core fucosylation are dramatically elevated in individuals with lung adenocarcinoma, promoting PD-1 expression and suppressing immune response to mediate tumorigenesis, growth, and metastasis [46]. Additionally, FUT8 also regulates PD-1 stability by controlling its degradation. By enhancing binding to FBXO38, a specific PD-1 E3 ubiquitin ligase that regulates PD-1 through Lys48-linked poly-ubiquitination, FUT8 ablation can enhance PD-1 ubiquitination [45] degradation by the 26S proteasome [47].

Consequently, Okada et al. elucidated that 2-fluoro-L-fucose (2F-Fuc) metabolically inhibits fucosylation, and can increase T-cell activation in particular and inhibit PD-1’s steady expression, providing attractive therapeutic opportunities targeting PD-1 [44]. In conclusion, core fucosylation and its associated components can hopefully become new biomarkers aiding PD-1 detection, patient stratification, and clinical treatment in the future.

### 3.2. Glycosylation-Specific Antibodies Targeting PD-1

Although anti-PD-1 antibodies have exhibited revolutionary clinical benefits against tumors, most patients do not respond completely or even show resistance to immune checkpoint inhibitors [48]. Hence, a more effective therapy blocking the PD-1 pathway is urgently needed.

Sun et al. generated a particular mAb against glycosylated PD-1 (gPD-1) called STM418, which potently neutralized interaction between PD-1 and PD-L1 and showed a significantly improved binding ability compared to traditional anti-PD-1 antibodies [49]. This is because STM418 primarily recognizes and functions on the N58 residue, the key glycosylated site mediating PD-1/PD-L1 interaction, while the binding interface of nivolumab or pembrolizumab is away from all four N-linked glycosylation sites and is hindered by glycan moieties [49,50]. STM418 also induced higher levels of IFN-γ and IL-2 secretion, together with enhanced T-cell proliferation and activation [49]. Moreover, STM418 increased stronger antitumor immunity by impairing tumor growth and decreasing tumor burden safely compared with nivolumab or pembrolizumab [49].

Camrelizumab is a humanized anti-PD-1 IgG4 mAb used to treat classical Hodgkin lymphoma that has relapsed or is resistant [51]. Camrelizumab’s blocking mechanisms involve binding between the heavy chain and PD-1, while the light chain sterically inhibits PD-L1 from binding to PD-1 [41]. Meanwhile, the N-glycans on the N58 residue of PD-1 are crucial for the affinity of camrelizumab with PD-1; therefore, the blocking effectiveness of camrelizumab for PD-1/PD-L1 is significantly decreased with PD-1 that lacks N58 glycosylation [41]. Taken together, antibodies targeting the glycosylation of PD-1 might substantially impress and revolutionize immunotherapy with its high efficacy and potential benefits.

SEA-TGT is an antibody targeting non-fucosylated T-cell immunoreceptors with Ig and ITIM domains (TIGIT) developed by sugar-engineered antibody (SEA) technology [52]. It can decrease T-cells with high TIGIT expression and reverse T-cell exhaustion; therefore, it is an ideal candidate for anti-PD-1 combination therapy. A phase I clinical trial has been proposed to test its safety and efficacy in advanced tumors.

## 4. Glycosylation Mediates PD-L1 and PD-1 Interaction

To conduct inhibitory signals to suppress T-cell activation, PD-1 has to interact with its ligand, PD-L1, and this process requires the assistance of glycosylation. B3GNT3, an N-glycosyltransferase in the Golgi apparatus, is in charge of attaching GlcNAc to core 1 in a β-1,3-linkage to generate extended core 1 oligosaccharides [53]. According to Li et al., B3GNT3, upregulated by EGF/EGFR signaling, acts as the catalyst for synthesizing polyLacNAc repeats presented on the glycan structure attached to both N192 and N200 residues of PD-L1 for improved physical contact with PD-1 [15]. This implies that glycosylation is necessary to regulate PD-1/PD-L1 interaction [49].

The α-mannosidase-II gene (MAN2A1) is an essential enzyme in the manufacture of complex N-glycans [54]. Shi et al. elucidated that MAN2A1 loss or inhibition changes the overall N-glycan composition of PD-L1 in cancer cells. With altered glycosylation patterns, PD-L1 cannot be well recognized by PD-1, therefore blocking their interactions. Shi et al. also discovered that the inactivation of MAN2A1 in cancer cells may make tumors more susceptible to anti-PD-L1 immunotherapy [55] (Figure 2). Therefore, MAN2A1 inhibitors are an ideal candidate for combination immunotherapy.

Taken together, N-linked glycosylation PD-L1 affects PD-1/PD-L1 interaction. Thus, glycosylation can be used as a new target for immune checkpoint inhibitors to improve the efficacy of immunotherapy.

## 5. The Glycosylation of PD-L2

Programmed death-ligand 2 (PD-L2) is a crucial immune checkpoint that has a vital status in T-cell malfunction and immune evasion. Its binding affinity is similarly controlled by glycosylation. Thus, understanding the molecular process and biological role of PD-L2 glycosylation will help develop new cancer immunotherapy strategies.

Since PD-L2 is substantially N-glycosylated at residues N64, N157, N163, and N189, the glycosylation of these sites except for N64 stabilizes PD-L2, reduces the killing ability of T-cells to cancer cells, and performs an essential function in mediating PD-1/PD-L2 interaction [56]. Xu et al. discovered that EGF/EGFR signaling upregulates PD-L2 glycosylation through STAT3-dependent transcriptional activation of FUT8 in patients with head and neck squamous cell carcinoma (HNSCC) resistant to cetuximab; PD-L2 is thereby prevented from polyubiquitination and lysosomal degradation, increasing its membrane accumulation and PD-1 binding affinity. Furthermore, Wang et al. demonstrated that PD-L2 is also stabilized by glycosylation in colorectal cancer (CRC) cells so that deglycosylation can predominantly inhibit PD-L2 expression [57]. Taken together, PD-L2 has the potential to be a novel research object for greater comprehension of the impact glycosylation exerts on PD-L2 stabilization and its interaction with PD-1, therefore enabling researchers to develop specific and effective agents against this target.

## 6. The Glycosylation of B7-H3

B7 homolog 3 protein (B7-H3), also referred to as CD276, is a crucial immune checkpoint member of the B7 immunoglobin superfamily [58]. B7-H3 is expressed on immune cells, especially antigen-presenting cells, and contributes to T-cell mediated immune response. B7-H3, which is aberrantly expressed in a wide variety of malignancies, is linked to a poor prognosis as well as tumor proliferation, migration, evasion, and angiogenesis [59]. These findings led scientists to believe that B7-H3 might be a compelling immunotherapy treatment target.

B7-H3 is confirmed to be N-glycosylated, and it has eight N-glycan sites, which are separated into four pairs in each of the IgV-IgC domains: N91 and N309, N104 and N322, N189 and N407, and N215 and N433 [60]. In TNBC cells, non-glycosylated B7-H3 showed a higher turnover and degradation rate than glycosylated B7-H3, suggesting that non-glycosylated B7-H3 proteins show decreased stability and are more prone to degradation facilitated by proteasome-mediated ubiquitination, and this indicates that the stability of the B7-H3 protein and its cell expression are caused by N-glycosylation [60]. Glycosylated B7-H3 greatly inhibited T-cells from proliferation and activation, suggesting that N-glycosylation is involved in B7-H3-mediated immunosuppression. Because N-glycans of B7-H3 from Ca9-22 oral cancer cells have been shown to have higher amounts of fucosylation than normal cells [61], Huang et al. identified that direct core fucosylation by FUT8 maintains the stability of B7-H3 in TNBC cells and mediates immunosuppression, which is relevant for patients with TNBC with a poor prognosis [60,62]. Thus, B7-H3 expression can be significantly inhibited by 2F-Fuc, which restores cancer cells’ sensitivity to cytotoxic T-cell-mediated immune response and functions as a combination therapy to elevate the efficacy of PD-L1-targeting treatment [60].

## 7. The Glycosylation of B7-H4

B7-H4 is a member of the B7 immunoglobulin superfamily expressed on professional APC and widely distributed in nonlymphoid tissues [63,64,65]. With varying levels of overexpression on multiple cancers, B7-H4 is associated with a poor prognosis [66]. B7-H4 helps tumor cells avoid immune surveillance and prevents T-cell activation and cytokine production to suppress immune response [63,64].

In TNBC, Song et al. unraveled that B7-H4 mainly has three glycosylation forms and five N-glycosylation sites (N112, N140, N156, N160, and N255). It has been proposed that glycosylation increases B7-H4 protein stability, since the B7-H4 variants that are unglycosylated and less glycosylated exhibited a greater turnover rate than the highly glycosylated version [67]. After using MG132, which specifically inhibits proteasome, both the ubiquitin conjugates of B7-H4 and its non-glycosylated form were primarily more abundant, indicating that the glycosylation of B7-H4 maintains its stability through the ubiquitin-proteasome pathway [67]. A variety of molecules, such as glycoprotein glucosyltransferase (UGGG1) and several OST complex subunits like RPN1, RPN2, and STT3A, interact with the autocrine motility factor receptor (AMFR), an essential E3 ubiquitin ligase that allows B7-H4 to take part in ERAD [67,68]. Furthermore, by suppressing the phosphorylation of eukaryotic translation initiation factor 2 subunit alpha (eIF2a) required for calreticulin exposure, B7-H4 expression can inhibit cell death induced by doxorubicin [67]. NGI-1 is an OST inhibitor that can inhibit the activities of STT3A and STT3B, subsequently decreasing B7-H4 glycosylation that improves the immunogenicity of cancer cells treated with doxorubicin. In preclinical TNBC models, NGI-1, camsirubicin (a non-cardiotoxic doxorubicin analog), and the PD-L1 blockade are used as a triple combination that effectively reduces tumor growth. Taken together, targeting B7-H4 glycosylation can improve the immunogenic properties of immune-cold cancers, and therefore facilitate other immunotherapeutic agents to promote treatment efficacy.

## 8. The Glycosylation of CTLA-4

CTLA-4, also known as CD152, competitively binds to CD80 (B7-1)/CD86 (B7-2) with a stronger binding affinity compared with its homolog CD28 and negatively regulates T-cell activation [69]. Unlike PD-1 primarily located at the plasma membrane, CTLA-4 is mainly located in intracellular vesicles. Once the T-cell is activated, CTLA-4 will translocate to the cell surface and play its immunosuppressive function [70].

N-glycosylation plays an important role in CTLA-4 internalization [71]. CTLA-4 has insufficient N-glycosylation caused by a T17A polymorphism in the signal peptide of CTLA-4, which decreases CTLA-4 surface delivery [72]. In multiple sclerosis, N-acetylglucosaminyltransferase I (Mgat1) expression and N-glycan branching are enhanced by vitamin D_3_ treatment, therefore inhibiting CTLA-4 internalization and increasing the surface expression of CTLA-4 in T-cells [73]. Besides, hexosamine metabolism and N-glycan branching increased by TCR signaling also influence CTLA-4 glycosylation and surface expression positively [74].

Abatacept is a CTLA-4-Ig that is highly glycosylated and has therapeutic functions on rheumatoid arthritis (RA) [75]. Zhu et al. used liquid chromatography-mass spectrometry (LC-MS) to characterize the N- and O-glycosylation modification of CTLA-4-Ig, which brings benefits for future studies addressing the similarity of CTLA-4-Ig biosimilars.

## 9. Sialylation and Siglecs

The sialic acid-binding immunoglobulin-like lectins (siglecs) belong to immunoglobulin-type lectins, which specifically recognize and bind to sialoglycans to regulate immune response [69,76] (Figure 3). Fifteen members of siglecs from Siglec-1 to Siglec-16 (except Siglec-13) can be divided into two subgroups: a group comprising Siglec-1, 2 (CD22), 4, and 15, and the Siglec-3 (CD33)-related group [77,78]. Siglecs are mostly expressed on the immune cell surface, where they function as immune response antagonists by inhibiting cytoplasmic immunoreceptor tyrosine-based (ITIM) motifs. Siglecs influence intracellular signaling by engaging SHP1 and SHP2 phosphatases, similar to other inhibitory receptors like PD-1. However, a significant distinction between siglecs and other inhibitory receptors is their ability to interact with both cis and trans ligands [79] (Figure 3).

Sialoglycans significantly enhanced their expression in tumor microenvironments due to hypersialylation, which results in the production of ligands for inhibitory siglec receptors on immune cells [80]. For example, pancreatic ductal adenocarcinoma (PDAC) cells exhibit increased sialylation regulated by sialyltransferases ST3GAL1 and ST3GAL4 that can be recognized by Siglec-7 and Siglec-9 on myeloid cells, which dictate monocyte to macrophage differentiation associated with poor clinical outcomes [81]. These results indicate that siglecs and their sialoglycan ligands are quite essential in multiple aspects of the biological processes of cancer cells, and they shed light on explorations of their possibilities to become targets for improved immunotherapy by understanding the mechanisms of their immune modulation ability.

### 9.1. Interaction between Siglecs and Its Sialoglycan Ligands

Sialic acids are presented on all cells, so they can be identified as hallmarks of ‘self’. However, some pathogens mimic sialoglycan ligands by coating themselves with sialylation, so siglecs will recognize them as ‘self’ ligands and therefore downregulate immune cell response to establish an immunosuppressive microenvironment [77]. For instance, Siglec-7 and -9 are inhibitory receptors on human NK cells independent from MHC I. Jandus et al. found that tumor cells of various histological types expressed the ligands of Siglec-7 and -9, and their interaction with Siglec-7 and -9 was linked to the susceptibility to NK cell-mediated cytotoxicity [82]. Normally, siglecs have deficient expression on human T-cells. Still, in contrast, Siglec-9 showed unexpected upregulation on TILs from colorectal cancer, ovarian cancer, melanoma, and patients with NSCLC, and its interaction with its ligands predominantly suppressed TCR-mediated activation and CD8^+^T-cell effector functions [83,84]. In addition, myeloid cells express a diverse panel of siglecs. Via the interaction between Siglec-9 and cancer-specific sialylation, in addition to inducing a tumor-associated macrophage (TAM)-like phenotype linked to elevated PD-L1 expression, the mucin MUC1 also caused calcium flow, which activated the MEK–ERK kinases in macrophages [85]. In ovarian and breast cancer, tumor-expressed CD24 can interact with Siglec-10 expressed by tumor-associated macrophages to inhibit tumor cell phagocytosis and promote immune evasion [86]. Taken together, the interaction between siglecs and its sialoglycan ligands can be considered a potential target for improving inhibitory immune response caused by siglec receptors.

### 9.2. Agents Targeting Sialoglycan–Siglec Immune Checkpoint

Siglecs are becoming more and more recognized as potential targets for immunotherapy [87]. Since siglec-specific antibodies have been used as agents for leukemia and lymphoma for more than 20 years, there is a plethora of evidence of their use.

Siglec-2 and Siglec-3 are both highly expressed in certain types of lymphoma and leukemia, which enables them to be direct targets for various treatments like antibodies–drug conjugates (ADCs) [88] or chimeric antigen receptors (CARs) [89] (Figure 3). For example, inotuzumab ozogamicin, an anti-Siglec-2 antibody conjugated to calicheamicin, resulted in a higher complete remission rate and improved progression-free survival (PFS) and overall survival (OS) than standard therapy in relapsed acute lymphoblastic leukemia [88]. A phase I clinical trial dual-targeted Siglec-2 and CD19 to generate a bispecific CAR for patients with large B cell lymphoma (LBCL) or relapsed/refractory B cell acute lymphoblastic leukemia (B-ALL) [89]. Gemtuzumab ozogamicin, an anti-CD33 antibody conjugate, is added in fractionated lower doses to the conventional therapy regimen for patients with acute myeloid leukemia. This allows for the safe administration of higher cumulative doses and significantly enhances patient outcomes [90].

Targeting sialoglycan ligands directly is another method to disrupt the linkage of siglecs with sialoglycan ligands. Agents such as sialic acid mimetic or sialyltransferase inhibitors can be exploited to interfere with sialic acid biosynthesis, thus reducing hypersialylation and inhibiting tumor immune evasion (Figure 3). Through intratumoral injection, the sialic acid mimic Ac53FaxNeu5Ac could inhibit tumor cells from expressing sialic acid and reduce tumor proliferation dependent on CD8^+^ T-cells in several tumor types [91]. Xiao et al. developed an antibody–sialidase conjugate by chemically fusing a recombinant sialidase to the human HER2-specific antibody trastuzumab. By selectively desialylating the tumor cell, bindings to inhibitory siglecs on NK cells were reduced in a HER2-dependent way to increase tumor cell vulnerability to ADCC [92].

Despite taking siglecs and sialoglycan ligands as direct tumor antigens, blocking interactions between them by inhibitory antibodies could restore immune response (Figure 3). Ibarlucea-Benitez et al. generated and identified that anti-Siglec-7 and anti-Siglec-9 antibodies, with their Fc regions modified to prevent engagement of Fcγ receptor, significantly decreased tumor burden and enhanced antitumor immunity [93]. There have been reports of interactions between Siglec-10 and the highly sialylated version of CD24, therefore Barkal et al. discovered that antibodies against CD24–Siglec-10 interaction increased all CD24-expressing cancer cell lines’ phagocytosis, indicating the clinical potential of this specific immune checkpoint blockade in cancer immunotherapy [86]. Human tumor cells and TAMs have high concentrations of Siglec-15, a molecule that suppresses macrophage-associated T-cells [94]. Wang et al. generated a monoclonal antibody specifically targeting Siglec-15 (α-S15, clone 5G12), which promoted CD8+ T-cells’ proliferation and suppressed tumor growth in mice, indicating that mAb acting against Siglec-15 has great therapeutic potential for immunotherapy [94].

## 10. Challenges and Future Perspectives

In recent years, cancer immunotherapy, especially mAbs targeting immune checkpoint molecules, has become a vital part of clinical use. However, due to the unsatisfying response rate, glycosylation, one of the most important PTMs, has gained attention for its crucial role in maintaining protein stability and mediating protein–protein interaction [95]. Despite the progress achieved in specific antibodies and small molecular agents targeting glycosylation, some limitations and challenges are also emerging: (1) Although deglycosylation promotes protein degradation, it might also negatively influence treatment efficacy. Cemiplimab is an anti-PD-1 antibody approved for locally advanced or metastatic cutaneous squamous cell carcinoma. Unlike nivolumab or pembrolizumab, whose treatment functions are independent of N-glycosylation, the binding affinity of cemiplimab is significantly decreased when the N58-glycan of PD-1 is removed, therefore impairing its antitumor efficacy [96]. (2) Glycosylation modification is rather complex and diverse; the accurate identification of glycosylation sites and glycan composition requires further exploration. With the development in mass spectrometry instrumentation, a more comprehensive understanding of immune checkpoint glycosylation can be reached through multi-omics analysis. (3) The molecular mechanisms underlying tumor glycosylation remain to be thoroughly explored, such as glycosylation-mediated immune evasion. The upstream and downstream molecules also have the potential to become novel drug targets. (4) The translation from basic research results to clinical practice is still on the way. Some preclinical and clinical trials testing glycosylation-specific drugs are ongoing, but the safety and efficacy remain to be seen. Hopefully, with the multidisciplinary cooperation among immunologists, biologists, bioinformaticians, and clinicians, the glucosylation of immune checkpoint proteins can be thoroughly elucidated from basic molecular structures, biological functions, and mechanisms, to clinical translations that bring benefits to patients with tumors.

## 11. Conclusions

Immune checkpoint inhibitors have shown remarkable success in the fight against a variety of tumors during the past few decades. Anti-PD-1/PD-L1 mAbs are the most widely explored and utilized agents of ICB that open up a new area of therapeutics. However, response rates still remain relatively low in most cancers, which calls on researchers to dig deeper to enhance the effectiveness of ICB. Glycosylation, as one of the most important types of PTM, is thoroughly discussed in this review to figure out its relationship with the stability of immune checkpoints and their interactions with ligands or receptors, together with the possibilities of its application in clinical aspects in the coming future. In addition to the immunosuppressive function, glycosylation also acts as a steric hindrance of antibody detection, as glycan moieties cover the binding site of antibodies. As a result, deglycosylation is an ideal treatment to enhance the sensitivity of detection, which can better guide patient stratification and subsequent treatment. Deglycosylation can also be clinically useful in treatment since it prevents glycosylation from stabilizing immune checkpoints, which causes subsequent degradation. On the other hand, some small-molecule agents targeting glycosylated immune checkpoints provide a brand-new perspective for us. Glycosylation-specific antibodies that recognize certain glycosylation sites can specifically bind to glycosylated immune checkpoints, disrupting the interaction between receptors and ligands effectively, and profoundly enhancing immune response by inhibiting the transduction of inhibitory signals. Other agents interfere with glycosylation composition through synthesis and metabolism pathways, affecting various immune checkpoints’ biological functions, and restoring host immunity against tumor cells (Table 1). Siglecs are novel immune checkpoints that might bring new insights into immunotherapy and require further in-depth investigation. Considering its similarity of modulation of intracellular signaling and its interaction with its sialoglycan ligands with other inhibitory immune checkpoints like PD-1, research ideas and methods of other well-established immune checkpoints can be used for reference.

## Figures and Tables

**Figure 1 biomedicines-12-01446-f001:**
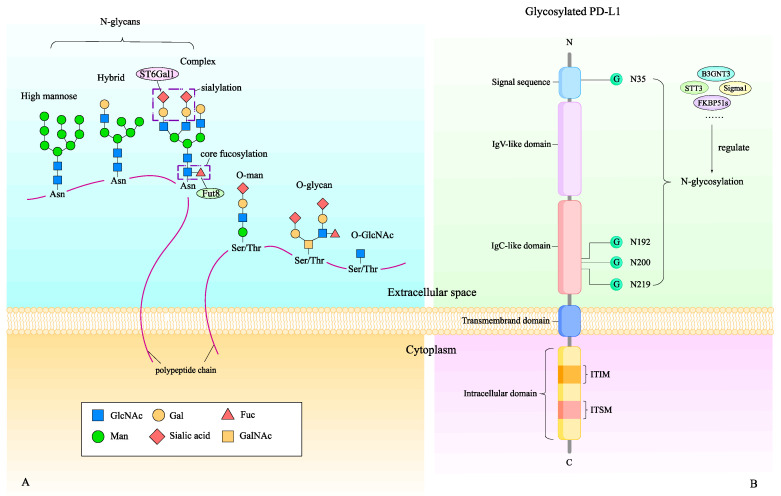
Common classes of glycosylation and the structure of PD-L1. (**A**) Depending on the different linkages formed with certain asparagine (Asn) or serine/threonine (Ser/Thr) residues, respectively, on the polypeptide backbone, glycosylation can be mainly divided into two types: N-linked glycosylation and O-linked glycosylation. N-glycans can be further classified into high mannose, hybrid, and complex types, which share common core regions but are modified by unique terminal structures to distinguish themselves from each other. N-acetylgalactosamine (GalNAc), O-linked to Ser/Thr, initiates mucin-type O-glycans, which are typically seen in secreted or membrane-associated glycoproteins. Other types of O-glycans include O-mannose (O-Man), O-linked β-N-acetylglucosamine (O-GlcNAc), and so on. (**B**) PD-L1 is a type 1 transmembrane protein composed of a signal sequence, two extracellular Ig-like domains, a transmembrane (TM) region, and an intracellular domain (ICD). The molecular weight of glycosylated PD-L1 increased to 50 kDa from 33 kDa, due to the addition of a 17 kDa N-glycan moiety. PD-L1 glycosylation regulated by multiple mechanisms occurs on its extracellular domain, and the four N-glycosylation sites are N35, N192, N200, and N219.

**Figure 2 biomedicines-12-01446-f002:**
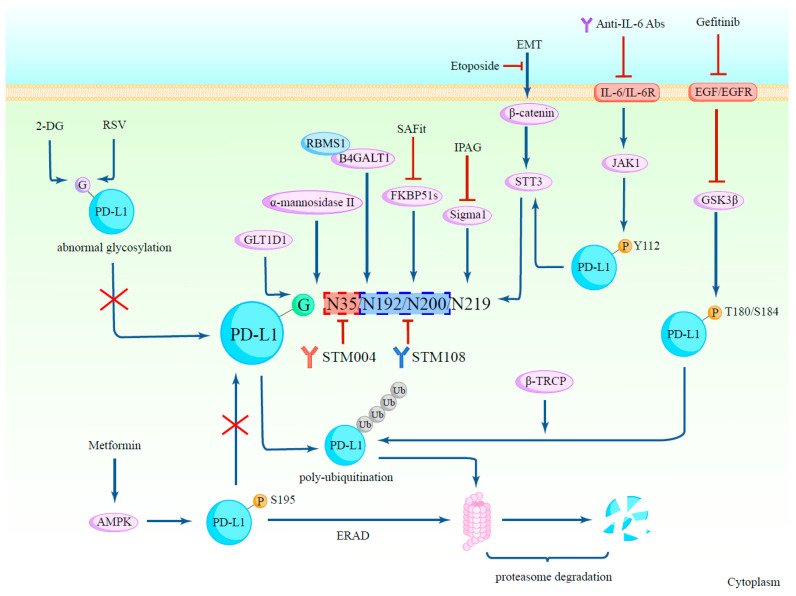
Mechanisms regulating PD-L1 N-linked glycosylation and small molecule therapeutics. The process of PD-L1 N-linked glycosylation is regulated by multiple mechanisms to maintain its protein stability. The green circle represents glycosylation, while the purple circle represents abnormal glycosylation, and the orange circles represent phosphorylation. Molecules that mainly regulate PD-L1 glycosylation are in purple ellipses. A detailed description of each pathway is included in the main text.

**Figure 3 biomedicines-12-01446-f003:**
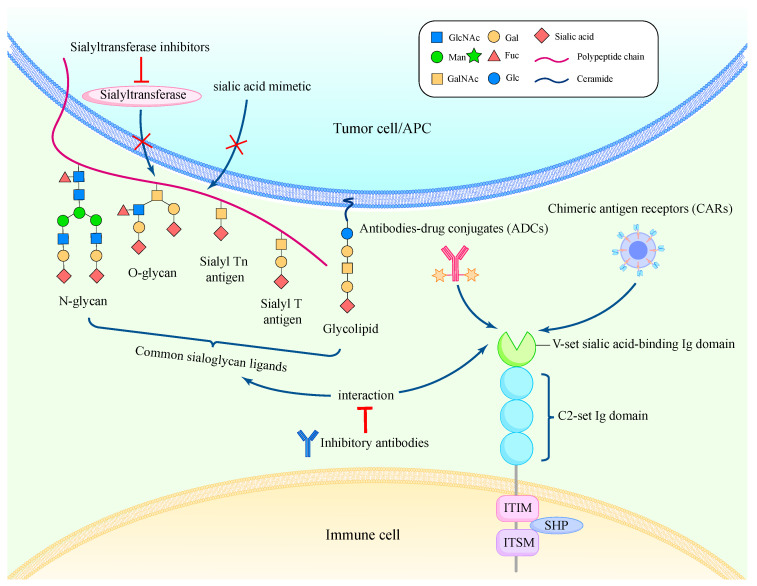
Structures of siglecs and their sialoglycan ligands and strategies targeting them. Siglecs are type 1 membrane proteins containing an amino-terminal V-set immunoglobulin domain that interacts with sialoglycan ligands and varying numbers of C2-set immunoglobulin domains. Common sialoglycan ligands include glycoproteins and glycolipids modified by various terminal structures. A wide range of therapeutics targeting the siglec–sialoglycan ligand axis has been explored and applied to improve the efficacy of immunotherapy.

**Table 1 biomedicines-12-01446-t001:** Summary of immune checkpoint proteins and related glycosylation.

Immune Checkpoints	Glycosylation Sites	Related Enzymes	Biological Effects	Potential Clinical Applications
PD-L1	N35, N192, N200, N219	STT3 Sigma1 FKBP51s B4GALT1 GLT1D1 B3GNT3 GSK3β	protect PD-L1 from degradation and enhance its protein stability facilitate binding with PD-1 and mediate immunosuppressive function	biomarker for clinical diagnosis and PD-L1 detection small molecular agents targeting PD-L1 glycosylation can improve antitumor immunity and can be used as combination therapy
PD-1	N49, N58, N74, N116	B3GNT2 FUT8	upregulate PD-1 expression and protein stability	the fucosylation inhibitor can increase T-cell activation and inhibit PD-1 expression glycosylation-specific antibodies can inhibit PD-1/PD-L1 interaction and increase treatment efficacy
PD-L2	N64, N157, N163, N189	FUT8	stabilize PD-L2 and mediate PD-1/PD-L2 interaction	have the potential to become a novel therapeutic target
B7-H3	N91, N309, N104, N322, N189, N407, N215, N433	FUT8	stabilize B7-H3 and mediate immunosuppressive function	use fucosylation inhibitors to restore cancer cells’ sensitivity to immune response and functions as a combination agent with immunotherapy
B7-H4	N112, N140, N156, N160, N255	UGGG1 RPN1/2 STT3A	stabilize the structure of the B7-H4 protein inhibit the immunogenicity of cancer cells	an OST inhibitor can be used as a triple combination with camsirubicin and the PD-L1 blockade to suppress tumor growth in TNBC
CTLA-4	N113, N145	Mgat1	inhibit CTLA-4 internalization and surface expression	is related to autoimmune diseases and its treatment

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
