# Peer review of "Glycosylation and Its Role in Immune Checkpoint Proteins: From Molecular Mechanisms to Clinical Implications"

_biomedicines, 2024, doi:10.3390/biomedicines12071446_

Round 1

Reviewer 1 Report

Comments and Suggestions for Authors

The review provides a comprehensive overview of the role of glycosylation in immune checkpoint proteins, highlighting its significance from molecular mechanisms to clinical implications in cancer immunotherapy. The authors effectively discuss how glycosylation influences protein stability, interaction with ligands, and therapeutic responses, providing insights into potential strategies to enhance immunotherapy effectiveness.

However, it would be helpful for the authors to further report specific examples or case studies where targeting glycosylation has shown promise in improving clinical outcomes.

In addition to the Figures, it is crucial to incorporate a table that offers an organized summary of the text's data. The table could feature columns for immune checkpoint proteins, specific glycosylation sites, enzymes involved in glycosylation, effects on protein function/stability, and potential clinical applications. This would offer readers a clear and concise overview of the key molecules and their roles within the context of immune checkpoint modulation.

Moreover, the authors should consider including a separate section that discusses the challenges, limitations and future directions in this field, providing a more balanced perspective. This could involve addressing unresolved questions, the development of more targeted glycosylation-modifying therapies, and the potential for personalized medicine approaches based on glycosylation patterns. Including information about the safety profile and potential side effects of glycosylation-modifying therapies could enhance their clinical utility.

Finally consider discussing the interdisciplinary nature of glycosylation research, including collaborations between immunologists, biologists, clinicians, and bioinformaticians. This could highlight the diverse expertise required to advance understanding and translation in this field.

Comments on the Quality of English Language

minor editing

Author Response

Dear reviewer:

Thank you for your decision and constructive comments on our manuscript. We have carefully considered the suggestions and made some revisions. The yellow parts are the changes we made according to your suggestions. Revision notes are given as follows:

  1. We added several specific examples of agents targeting glycosylation in PD-1 (lines 307-311, 515-519), B7-H4 (lines 389-398), and CTLA-4 (lines 413-417).
  2. We added a comprehensive table covering several immune checkpoint proteins and their glycosylation, please see detailed information in Table 1.
  3. We added a Challenges and Future Perspectives section, includes limitations, challenges and potential future developments.
  4. We added multidisciplinary collaboration in lines 528-532 as suggested.

We have tried our best to improve the manuscript and appreciate your useful suggestions. We hope our correction will meet with approval.

Sincerely,

Bo Feng

17 June, 2024

Reviewer 2 Report

Comments and Suggestions for Authors

This is an interesting review on the role of glycosilation in the function of immune checkpoints, as well as on the impact of glycosilation in immune checkpoint detection and therapeutical applications.

Comments on the Quality of English Language

While this is a relevant review, the manuscript requires revision of the scientific English language, to improve its synthax and make the review more readable, and thus more useful for the Journal audience. Also, authors should revise the use of abbreviations, and make it consistent, as there are abbreviations that appear without spelling of the term (in the Abstract section) and unnecessary abbreviations of terms appearing only once in the text.

Author Response

Dear reviewer:

Thank you for your decision and constructive comments on our manuscript. We have carefully considered the suggestions and made some revisions. The yellow parts are the changes we made according to your suggestions. Revision notes are given as follows:

  1. We checked and identified all the abbreviations we missed in the manuscript.
  2. We have invited a native speaker to help us polish our manuscript to make it more readable. 

We have tried our best to improve the manuscript and appreciate your useful suggestions. We hope our correction will meet with approval.

Sincerely,

Bo Feng

15 June, 2024

Reviewer 3 Report

Comments and Suggestions for Authors

This short review describes selected aspects of glycosylation in some immune checkpoint proteins. It joins other recent reviews in this rapidly expanding research field; see references 1-3 below for examples, which should be cited. The emphasis of the review is on therapeutic and diagnostic exploitation of glycosylation in immune checkpoint therapy. Most of the text is concerned with PD-1 and its ligands PD-L1 and PD-L2, with a shorter section on siglecs and sialylation and brief mention of B7-H3 and B7-H4. Other immune checkpoint proteins are not mentioned. CTLA-4 appears in the abstract but nowhere else in the article; a section on this protein should be included, as a simple literature search finds a few papers worth citing.  

In addition there are some minor presentation problems:

1.       The symbolic representation of glycan structures used in Fig. 1 does not conform to the standard system ( https://www.ncbi.nlm.nih.gov/glycans/snfg.html ). Please redraw this figure using the established conventions. See Figure 1 of ref. 4 below for an example.  

2.        Though the abbreviations are usually defined, there are exceptions (for example TNBC, TIL). Please help the non-specialist by defining all the abbreviations.

1: Chakraborty M, Kaur J, Gunjan, Kathpalia M, Kaur N.

Clinical relevance of glycosylation in triple negative breast cancer: a review. Glycoconj J. 2024 Apr;41(2):79-91. doi: 10.1007/s10719-024-10151-0. Epub 2024 Apr 18. PMID: 38634956.

2: Xi X, Zhao W.

Anti-Tumor Potential of Post-Translational Modifications of PD-1. Curr Issues Mol Biol. 2024 Mar 6;46(3):2119-2132. doi: 10.3390/cimb46030136. PMID: 38534752; PMCID: PMC10968922.

3: Wang J, Wang Y, Jiang X, Xu M, Wang M, Wang R, Zheng B, Chen M, Ke Q, Long J.

Unleashing the power of immune checkpoints: Post-translational modification of novel molecules and clinical applications. Cancer Lett. 2024 Apr 28;588:216758. doi: 10.1016/j.canlet.2024.216758. Epub 2024 Feb 22. PMID: 38401885.

4: Zheng L, Yang Q, Li F, Zhu M, Yang H, Tan T, Wu B, Liu M, Xu C, Yin J, Cao C.

The Glycosylation of Immune Checkpoints and Their Applications in Oncology. Pharmaceuticals (Basel). 2022 Nov 23;15(12):1451. doi: 10.3390/ph15121451. PMID: 36558902; PMCID: PMC9783268.

Author Response

Dear reviewer:

Thank you for your decision and constructive comments on our manuscript. We have carefully considered the suggestions and made some revisions. The yellow parts are the changes we made according to your suggestions. Revision notes are given as follows:

  1. We added a section about CTLA-4 and Glycosylation as suggested. However, since CTLA-4 is an immune checkpoint that is less glycosylated, there is limited literature we can refer to. If you have any recommendations, please remind us.
  2. We changed the symbolic representation of glycan structures in Fig. 1 and Fig.3 according to the standard system you recommended.
  3. We checked and identified all the abbreviations we missed in the manuscript.
  4. We added the references you recommended in the appropriate locations of the text as references 39, 60, 66, 94. 

We have tried our best to improve the manuscript and appreciate your useful suggestions. We hope our correction will meet with approval.

Sincerely,

Bo Feng

17 June, 2024